# Derivation and Flight Test Validation of Maximum Rate of Climb during Takeoff for Fixed-Wing UAV Driven by Propeller Engine

**Katsumi Watanabe, Takuma Shibata and Masazumi Ueba \***

Graduate School of Engineering, Muroran Institute of Technology, Muroran 050-0071, Japan
\* Correspondence: ueba@muroran-it.ac.jp

**Abstract:** In recent years, the use of fixed-wing Unmanned Aerial Vehicles (UAVs) has expanded, and the use of fixed-wing UAVs is expected to expand due to their usefulness for long-range operations. Different from manned aircraft, no provision is required regarding climb angle at takeoff for fixed-wing UAVs. Therefore, fixed-wing UAVs can take off by taking advantage of their performance. In addition, propeller engines are the propulsion device currently used by most fixed-wing UAVs. However, the thrust force generated by a propeller engine decreases as its airspeed increases. In such circumstances, this paper describes how to derive a maximum rate of climb in which the characteristics of the propeller engine are taken into account, with the aim of reducing takeoff time by maximizing the rate of climb during takeoff. The derivation uses optimization problems with a dependency of the thrust force on the airspeed. After the derivation of the maximum rate of climb, we first checked whether the maximum rate of climb obtained for the mass system was feasible for takeoff at the rate of climb by using a 6-DOF flight simulation, and then confirmed its validity through flight experiments.

**Keywords:** fixed-wing UAV; propeller; maximum rate of climb; flight verification

## 1. Introduction

In recent years, the use of UAVs has increased in many fields, including agriculture and transportation. These UAVs can be categorized as fixed-wing UAVs or rotary-wing UAVs. Fixed-wing UAVs have some advantages over rotary-wing UAVs, from the viewpoint of better energy efficiency in flight, more payloads, and higher cruising speeds. For this reason, they are most effective when used for long flight distance. Therefore, they are expected to be used in fields such as observation and transportation.

As fixed-wing UAVs in those fields are generally used out of sight, they can fly autonomously for all flight modes such as takeoff, cruise, and landing. Of these flight modes, there are a few studies on the takeoff mode of fixed-wing UAVs. Some takeoff studies have been conducted on VTOL aircraft: a study on generating a takeoff trajectory with minimum power consumption for tilt-wing aircraft [1], a study on maximizing payload weight by simultaneously optimizing the conceptual aircraft design and takeoff trajectory [2], and a study on deriving the optimal takeoff trajectory for tilt-rotor aircraft [3]; others are part of studies of autonomous flight covering the entire flight, from takeoff to cruise and landing [4,5]. However, VTOL equipment in aircraft lead to an increase in aircraft weight, which reduces the aircraft's endurance range and cruising speed. On the other hand, studies of autonomous flight from takeoff to landing have not considered the optimization of parameter settings during takeoff [4] nor have they mentioned parameter settings [5].

On the other hand, in recent years, studies [6,7] have been conducted on takeoff with regard to manned fixed-wing aircraft. Study [6] evaluates the aerodynamic performance of channel wings that increase lift and contribute to a short takeoff, and study [7] estimates the takeoff performance of an aircraft equipped with a DEP (Distributed Electric Propulsion) blown wing that increases lift. However, those techniques in the above studies deal with

the improvement of takeoff performance by using new wings, and require hardware modifications or an increase in the amount of propulsion equipment.

Generally, takeoff performance is specified by runway distance and takeoff time. This study aims at minimizing the takeoff time so that the mission can be started quickly. However, it costs a lot to reduce the time required to run because modifications of engines and aircraft hardware are required. Therefore, this study focuses on the takeoff time among the above takeoff performance parameters, and deals with a climb phase during the takeoff.

Until now, no study has attempted to minimize takeoff time by optimizing the takeoff trajectory. A manned airplane generally climbs at a minimum climb angle during takeoff, even if the airplane is capable of climbing well above the minimum climb angle [8]. The climb angle is determined by the height of obstacles and other objects on the takeoff trajectory. However, fixed-wing UAVs can climb by making the best use of their capability instead of the provisions required for a manned airplane.

As for conventional studies on climb, there is a study that obtained the shortest time climb trajectory based on the concept of energy altitude [9], a study that obtained the maximum climb angle trajectory of a supersonic interceptor using the steepest descent method [10], a study that derived the optimal climb trajectory for a supersonic airplane with different numbers of state variables using sequential quadratic programming and investigated changes in the trajectory [11], and a study that obtained the shortest climb trajectory for a supersonic airplane with a large thrust-to-weight ratio using the steepest descent method [12]. However, these studies are characterized by the use of jet engines as the propulsion system for the target airplane. Moreover, it generally takes a lot of time to numerically calculate the optimal climb paths of the airplane, because the above studies dealt with a dynamic system. Therefore, in this study, assuming that the transition time from run to climb is short, the steady climb section, which occupies most of the takeoff time, is targeted and solved as a static problem to shorten the takeoff time.

Considering the above circumstances, this paper, aiming at reducing the takeoff time of an airplane without VTOL equipment, proposes a new method to realize a maximum rate of climb for fixed-wing UAVs driven by propeller engines. The method is based on an optimization problem with equal constraints, an example of which in the aerospace field is generally known as a maximum steady state rate of climb for airplanes [13]. In the example, the characteristics of the propeller engine are not taken into account.

The proposed technology in this study requires that the aerodynamic and thrust characteristics of the fixed-wing UAV should be known with sufficient accuracy in advance. If the aerodynamic characteristics are unclear, a sudden rotation and climb will lead to a stall in a real flight environment, so it is necessary to have a good understanding of the characteristics.

Propeller engines are often used as propulsion equipment for fixed-wing UAVs because they are cheaper and easier to handle than jet engines. However, the thrust generated by propeller engines varies according to the airspeed of the fixed-wing UAVs. It is not easy to theoretically obtain the characteristics of a propeller engine, i.e., they should be calculated by using detailed data on the propeller shape through wind tunnel tests [14]. Therefore, how to incorporate the propeller engine characteristics is very important and should be clarified.

The takeoff profile of a fixed-wing UAV is usually divided into three phases: run phase, rotation phase, and climb phase, as shown in Figure 1 [15]. In this paper, only a climb phase is dealt with in order to clarify the effectiveness of our method. First, the equations of equilibrium acting on the airplane during a steady climb are described in Section 2, followed by an explanation of how to derive the maximum rate of climb using optimization problems. The equation relating the thrust generated by the propeller engine to airspeed, the specifications of the airplane under consideration, and the calculation results of the maximum rate of climb are also described. Section 3 describes the results of a 6-DOF flight simulation conducted to confirm whether the maximum rate of climb obtained from the analysis of a mass system can be realized in an actual 6-DOF environment. The section also

describes the control system to achieve steady climb at the maximum rate of climb used in the 6-DOF flight simulation. Section 4 describes the criteria and experimental results of the steady climb in the flight experiments using the model airplane to confirm the validity of the maximum rate of climb, and finally Section 5 concludes the study.

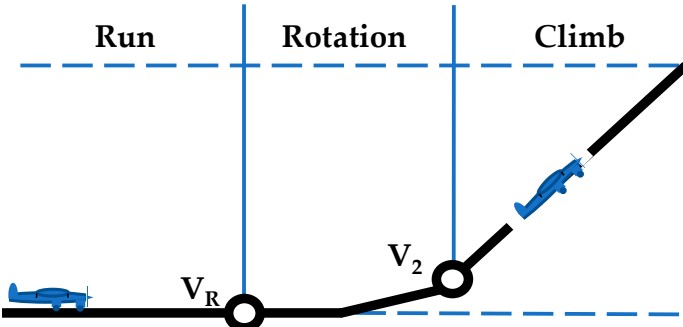

**Figure 1.** Takeoff profile.

## 2. Derivation of Maximum Rate of Climb Using Optimization Problems

### 2.1. Equations of Motion during Climb

Here, the motion of the airplane is expressed in the stable axis coordinate system, in which the x-axis fixed to the airplane is aligned with the direction of travel of the airplane. Furthermore, assuming the airplane keeps its wings horizontal and climbs straight along the centerline of the runway during takeoff, the lateral motion is neglected; therefore, only the longitudinal motion is considered in the optimization problem. The equations of longitudinal motion consist only of the translational motion of the x- and z-axes and the rotational motion around the y-axis. Moreover, we deal with the motion of the airplane in a steady state of climb, where the airplane is regarded as a mass point, and only the equilibrium Equations (1) and (2), which are obtained by decomposing the forces (lift, drag, gravity, and thrust), are acting on the airplane during climb, as shown in Figure 2, into the x- and z-axes of the stability axis.

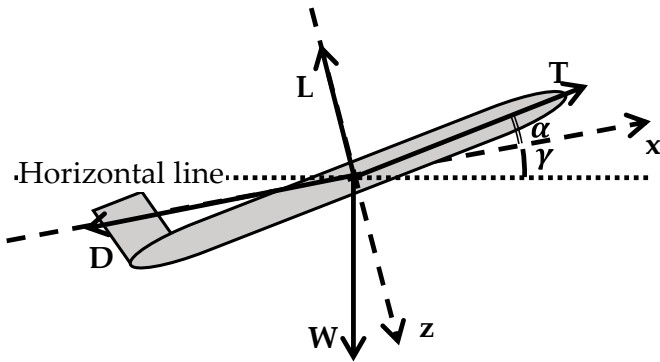

**Figure 2.** Forces acting on UAV during climb.

Equation of equilibrium in the x-axis of the stable x-axis:

$$T(V)\cos \alpha - D(V, \alpha) - W\sin \gamma = 0 \tag{1}$$

Equation of equilibrium in the z-axis of the stable y-axis:

$$T(V)\sin \alpha + L(V, \alpha) - W\cos \gamma = 0 \tag{2}$$

Lift and drag are defined as functions of the angle of attack $\alpha$ and airspeed V, and are expressed as Equations (3)–(5) [16].

$$L = \frac{1}{2}(C_{L0} + C_{L\alpha}\alpha)\rho SV^2 \tag{3}$$

$$D = \frac{1}{2}C_D\rho SV^2 \tag{4}$$

$$C_D = C_{D0} + \frac{(C_{L0} + C_{L\alpha}\alpha)^2}{\pi e AR} \tag{5}$$

### 2.2. Application of Optimization Problems to Climbing Airplane

Here, we apply an optimization problem [13] to derive the maximum rate of climb. The evaluation function J is a rate of climb as shown in Equation (6), and is to be maximized. The Hamiltonian H is defined as Equation (7) by using Equations (1) and (2) as constraints.

Variables in these equations are the airspeed V, the angle of attack $\alpha$, the path angle $\gamma$, and the adjoint variables $\lambda_1$, $\lambda_2$. The thrust T is a function of the airspeed V, and the lift L and the drag D are also a function of the airspeed V and the angle of attack $\alpha$.

$$J = V\sin\gamma \tag{6}$$

$$\begin{aligned} H = J(V,\gamma) &+ \lambda_1(T(V)\cos\alpha - D(V,\alpha) - W\sin\gamma) \\ &+ \lambda_2(T(V)\sin\alpha + L(V,\alpha) - W\cos\gamma) \end{aligned} \tag{7}$$

The necessary condition for maximizing the rate of increase, i.e., the evaluation function J (6), is that the partial derivatives for all functions of the Hamiltonian (7) are all zero. Therefore, we first obtain the partial derivatives (8) through (12) for all functions of the Hamiltonian. Next, we find the functions when all five derivatives are zero. As those equations are nonlinear and simultaneous with V, $\alpha$, $\gamma$, $\lambda_1$, and $\lambda_2$ as variables, they are to be solved numerically. Numerical analysis was performed using MATLAB software (version: R2020b) [17].

$$\frac{\partial H}{\partial V} = \frac{\partial J}{\partial V} + \lambda_1\left(\frac{\partial T}{\partial V}\cos\alpha - \frac{\partial D}{\partial V}\right) + \lambda_2\left(\frac{\partial T}{\partial V}\sin\alpha + \frac{\partial L}{\partial V}\right) = 0 \tag{8}$$

$$\frac{\partial H}{\partial \alpha} = \lambda_1\left(-T\sin\alpha - \frac{\partial D}{\partial \alpha}\right) + \lambda_2\left(T\cos\alpha + \frac{\partial L}{\partial \alpha}\right) = 0 \tag{9}$$

$$\frac{\partial H}{\partial \gamma} = \frac{\partial J}{\partial \gamma} - \lambda_1 W\cos\gamma + \lambda_2 W\sin\gamma = 0 \tag{10}$$

$$\frac{\partial H}{\partial \lambda_1} = T(V)\cos\alpha - D(V,\alpha) - W\sin\gamma = 0 \tag{11}$$

$$\frac{\partial H}{\partial \lambda_2} = T(V)\sin\alpha + L(V,\alpha) - W\cos\gamma = 0 \tag{12}$$

### 2.3. Formulation of Propeller Engine Thrust

There exist two theoretical methods for deriving the thrust of a propeller engine: one method is Rankine's momentum theory [18] and the other is the blade element theory [19]. Rankine's momentum theory regards a propeller as a single disk, and derives the thrust from the change in the momentum and energy of the air flowing into and out of this disk. This method does not take into account the propeller speed (rpm), the blade shape, or the number of blades that make up the propeller. The blade element theory, on the other hand, considers a propeller to be a collection of minute blades (blade elements), and derives the thrust of the propeller as a whole by integrating the forces acting on the blade elements over the entire propeller. However, to derive the thrust of the propeller using this method, it is necessary to accurately grasp the blade angle and airfoil shape. In addition, if the

propeller engine is to be experimentally characterized for thrust, facilities are needed to conduct wind tunnel tests. On the other hand, in order to clarify the thrust of the propeller using numerical analysis, detailed propeller geometry data are required, and it takes a lot of time to calculate the thrust.

For the above reasons, it is difficult to theoretically, experimentally, and numerically determine the thrust of a propeller engine. Therefore, this study uses airspeed vs. thrust data under constant propeller speed published by propeller manufacturers [20]. The thrust was derived for an engine with a propeller size of 14 inches by 8 pitch angles and a maximum propeller speed of 8000 rpm in the blue dot line shown in Figure 3. The relationship of airspeed with thrust was obtained as Equation (13) after a second-order approximation.

$$T(V) = -0.0167 \times V^2 - 0.497 \times V + 38.057 \tag{13}$$

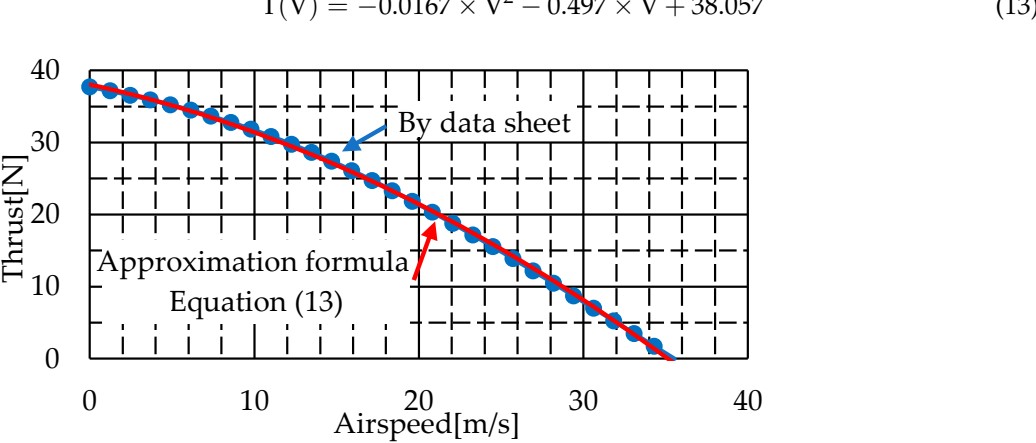

**Figure 3.** Relationship between propeller engine thrust and airspeed.

### 2.4. Target Airplane

The validity of the proposed method is confirmed by using a model airplane capable of horizontally flying at a maximum airspeed of about 35 m/s driven by propeller engines, as shown in Figure 4, which is a low-wing, front-wheel type of airplane equipped with guidance and control circuits and various sensors (Air Data Sensor (ADS), Inertial Navigation System (INS)). The specifications of this airplane are listed in Table 1. The 6-DOF flight simulation described in Section 3 requires the aerodynamic coefficient of the target airplane. There are several methods for calculating the aerodynamic coefficient, such as actual flight tests, wind tunnel tests, and the use of DATCOM (an aerodynamics analysis tool developed by the U.S. Air Force [21]). Among these methods, we used the estimation equation in reference [16] because the coefficient can be easily estimated. This method can estimate the coefficients of the airplane by using various dimensions of the airplane. Since the derivation methods for the coefficient of lift $C_{L0}$ and the coefficient of drag $C_{D0}$ are unknown for this method, the same values as those used for the airplane in reference [22] were used. These aerodynamic coefficients are shown in Table 2. The air density $\rho$ and gravitational acceleration g used in the calculations are also listed in Table 3.

**Table 1.** Specifications of model airplane.

| Airplane Specification | Value |
|:---:|:---:|
| m | 6.0 kg |
| c | 0.315 m |
| $\alpha_{stall}$ | 10 deg. |
| S | 0.649 m$^2$ |
| e | 0.6 |
| AR | 6.54 |

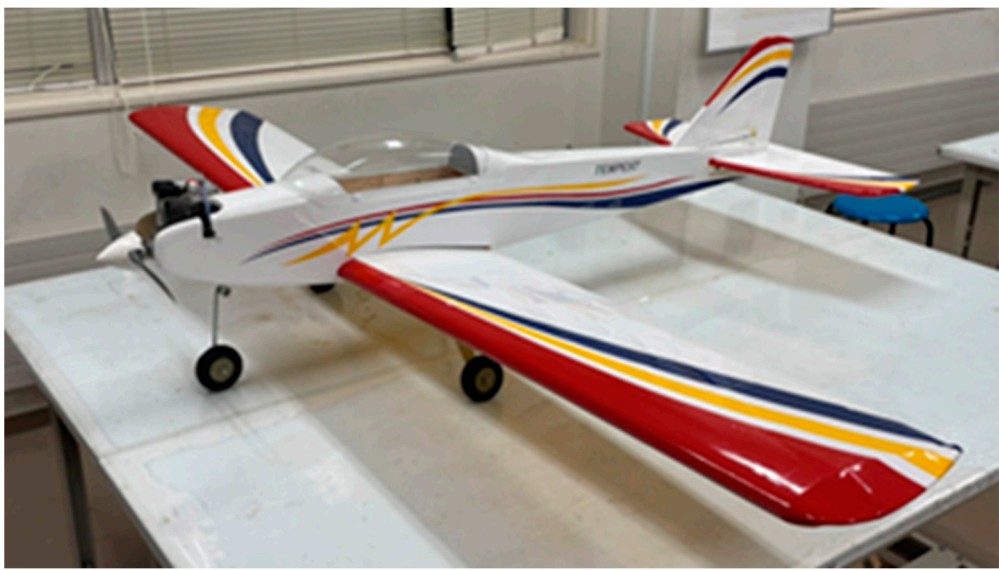

**Figure 4.** Appearance of model airplane.

**Table 2.** Aerodynamic parameters of model airplane.

| Longitudinal | | Lateral | |
|---|---|---|---|
| $C_{L\alpha}$ | 4.355 | $C_{y\beta}$ | −0.1623 |
| $C_{L0}$ | 0.176 | $C_{y_p}$ | −0.03861 |
| $C_{D0}$ | 0.0488 | $C_{y_{\delta_r}}$ | 0.08116 |
| $C_{xu}$ | −0.3790 | $C_{y_r}$ | 0.1431 |
| $C_{x\alpha}$ | 0.06371 | $C_{l\beta}$ | −0.05330 |
| $C_{zu}$ | 0 | $C_{l_{\delta_a}}$ | −0.4022 |
| $C_{z\alpha}$ | −4.355 | $C_{l_{\delta_r}}$ | 0.009653 |
| $C_{z_{\delta_e}}$ | −0.4324 | $C_{lp}$ | −0.7798 |
| $C_{zq}$ | −4.881 | $C_{lr}$ | 0.08940 |
| $C_{mu}$ | 0 | $C_{n\beta}$ | 0.04465 |
| $C_{m\alpha}$ | −1.359 | $C_{n_{\delta_a}}$ | 0 |
| $C_{m_{\delta_e}}$ | −1.220 | $C_{n_{\delta_r}}$ | −0.03578 |
| $C_{mq}$ | −13.78 | $C_{np}$ | −0.01062 |
| $C_{m\dot{\alpha}}$ | −5.111 | $C_{nr}$ | −0.1052 |

**Table 3.** Physical constants.

| Item | Value |
|---|---|
| $\rho$ | 1.23 kg/m$^3$ |
| g | 9.81 m/s$^2$ |

*2.5. Numerical Result of Maximum Rate of Climb*

By solving Equations (8)–(12) using the specifications shown in Tables 1 and 2, the maximum rate of climb was obtained, as shown in Table 4, with an airspeed of 15.1 m/s, an angle of attack of 5.4 degrees, and a path angle of 19.6 degrees. From these results, the pitch angle of the target airplane climbing in steady state was 24.9 degrees, and the maximum rate of climb was 5.1 m/s.

**Table 4.** Solutions of maximum rate of climb and flight variables.

| Flight Variable | Optimal Solution |
|:---:|:---:|
| V | 15.1 m/s |
| $\alpha$ | 5.4 deg. |
| $\gamma$ | 19.6 deg. |
| $\theta$ | 24.9 deg. |
| $V\sin\gamma$ | 5.1 m/s |

## 3. Verification by Using 6-DOF Flight Simulation

To confirm whether the flight at the maximum rate of climb derived on a mass system in the previous section is feasible in the model airplane, simulations from run to rotation and climb were carried out by using a 6-DOF flight simulator developed in-house in MATLAB and the Simulink program [23].

### 3.1. Simulation Condition

The speed at which the airplane transits from the run phase to the rotation phase (rotation speed $V_R$) was set to be equal to the stall speed ($V_{s1}$) in the takeoff configuration obtained from Equation (14), as in reference [22]. The switching speed from the rotation phase to the climb phase (safety takeoff speed $V_2$) was set to be 1.2 times the stall speed from reference [24], and these values are shown in Table 5. In addition, the simulation time was set to be 20 s by using the result of the takeoff of flight experiment in reference [4] in which the UAV used flew at a lower speed.

$$V_{s1} = \sqrt{\frac{2W}{\rho S C_{Lmax}}} \tag{14}$$

**Table 5.** Speed of phase switching.

| Flight Variable | Airspeed |
|:---:|:---:|
| $V_R$ | 12.5 m/s |
| $V_2$ | 15.1 m/s |

### 3.2. Flight Control System for Model Airplane during Takeoff

In this simulation, the entire takeoff process, including run, rotation, and climb, is verified. Therefore, three control systems (run control system, rotation control system, and climb control system) are required. The block diagram of the run control system is shown in Figure 5. The lateral-directional control system controls the steering so that the airplane's lateral position keeps along the centerline of the runway. The longitudinal system controls the elevator so that the steering gear does not leave the ground, i.e., the pitch angle is kept at 0 degrees.

After the airplane's airspeed reaches the rotation speed ($V_R$), the airplane transits from the run phase to the rotation phase. The block diagram of the rotation control system is shown in Figure 6. In the lateral-directional control system, the ailerons are controlled to keep the airplane horizontal. In the longitudinal control system, the nose is raised to a designated angle to increase the lift until the airplane leaves the ground. The pitch angle during this phase is set to be 10 degrees with 2 degrees of margin to prevent the tail of the airplane from hitting the ground during the run.

After reaching a safe takeoff speed ($V_2$), the airplane transits from the rotation phase to the climb phase. The block diagram of the climb control system is shown in Figure 7. During the climb phase, in the lateral-directional control system, the airplane climbs straight onto the runway, i.e., the lateral position of the airplane is maintained to be on the centerline of

the runway by controlling the ailerons. The longitudinal control system aims at achieving a steady climb at the maximum rate of climb. During this steady climb, the airplane's motion is balanced; if one of the airspeed, path angle, and angle of attack is determined, the remaining two values are uniquely determined. Therefore, the pitch angle is controlled with maximum thrust so that the airspeed matches the one (15.1 m/s) at which the maximum rate of climb can be achieved. When the airplane's airspeed is greater than that of the optimal solution, the pitch angle is increased so as to convert kinetic energy to potential energy. On the other hand, if the airplane's airspeed is below the optimal solution, the pitch angle is reduced so as to convert potential energy to kinetic energy. Here, the pitch angle of the optimal solution is added to the pitch angle command at the start of the climb phase. Furthermore, each control system uses PID controllers. The throttle is fixed at position where it generates maximum thrust during takeoff. However, the actual magnitude of the thrust in this case follows Equation (13). The airspeed used in the above flight control is measured by the ADS, and the attitude angle and position by the INS.

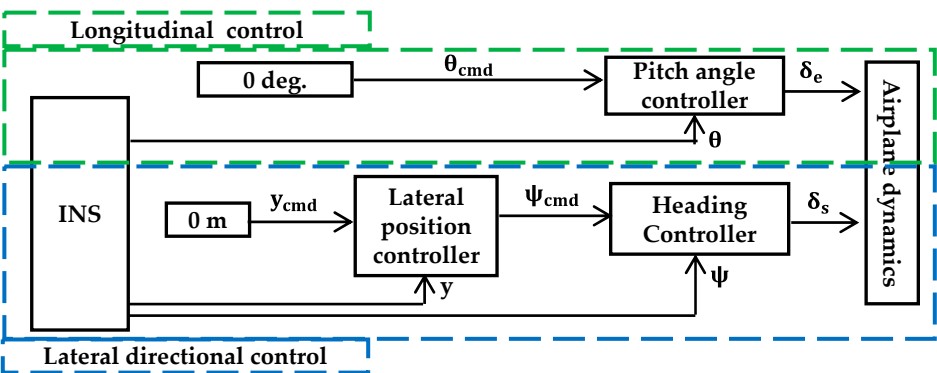

**Figure 5.** Block diagram of run control.

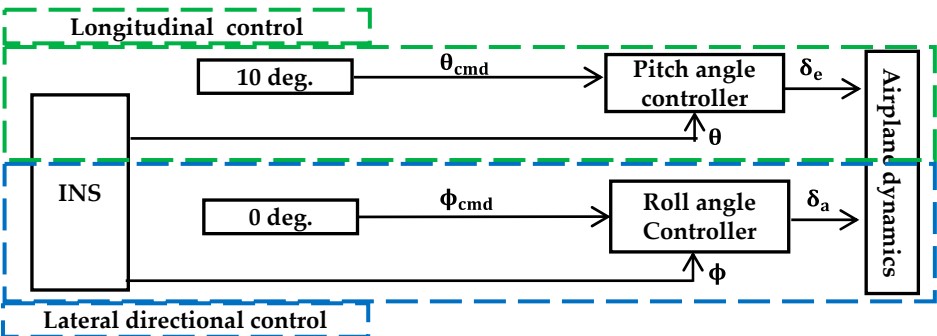

**Figure 6.** Block diagram of rotation control.

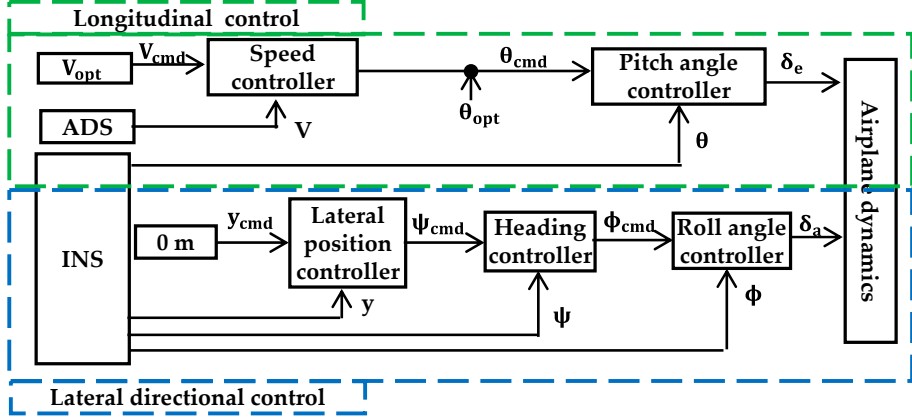

**Figure 7.** Block diagram of climb control.

### 3.3. Results of 6-DOF Flight Simulation

The simulation results for the entire takeoff, from run to rotation and climb, are shown in Figures 8–12. The airplane took off straight along the centerline of the runway without moving laterally, as shown in Figure 8. As shown in Figure 9, the airplane climbed to an altitude of about 80 m at the end of the simulation. Figures 10 and 11 show that the profile of the airspeed and the pitch angle exceeded the command values just after the start of climb, but in 3 s, the airspeed converged to 15.1 m/s and the pitch angle to 24.9 degrees, respectively. That is to say, the airplane reached a steady state of climb in about 7 s after the start of climb. The rate of climb is confirmed to be 4.9 m/s, as shown in Figure 12. The difference between the rate by simulation and the one by the optimal method is just 0.2 m/s, which is approximately 3.9% of the maximum rate of climb and negligibly small. Therefore, it was confirmed that the takeoff, including the climb at a maximum rate of climb, could be well achieved.

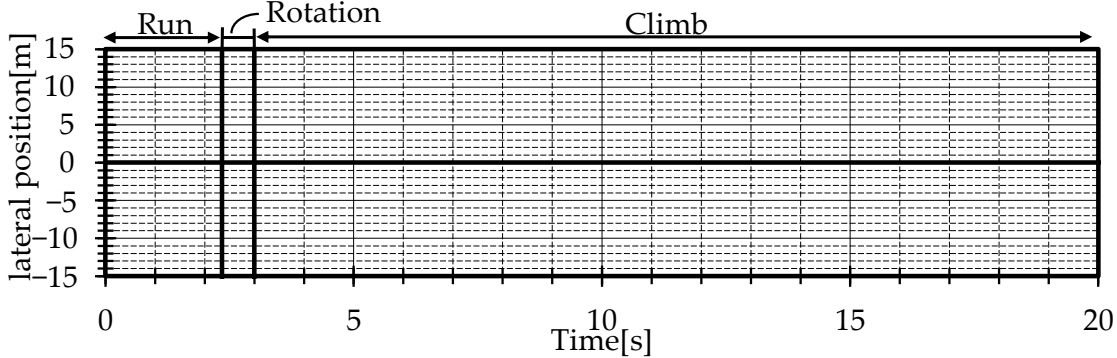

**Figure 8.** Simulated profile of lateral position.

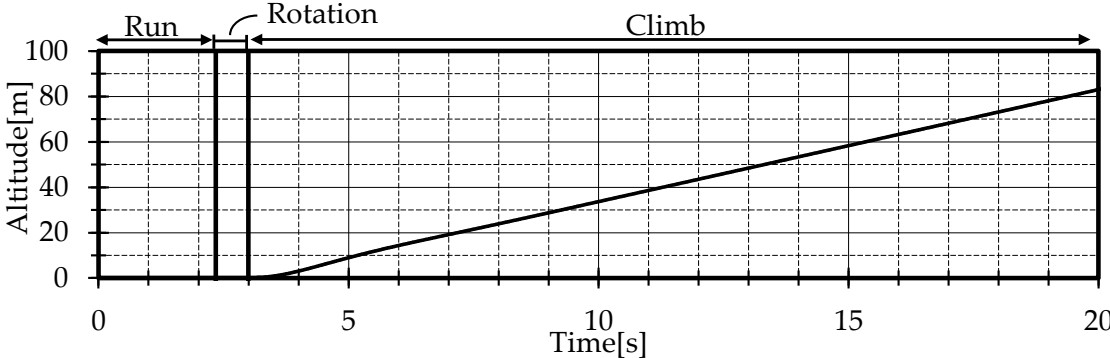

**Figure 9.** Simulated profile of altitude.

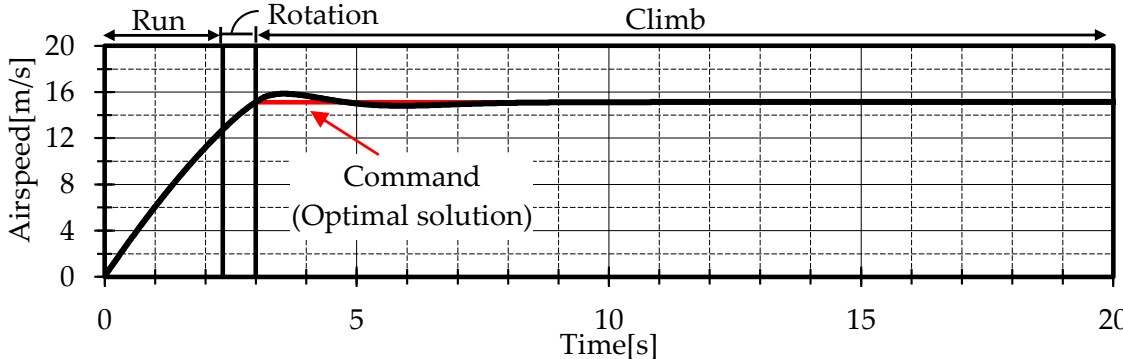

**Figure 10.** Simulated profile of airspeed.

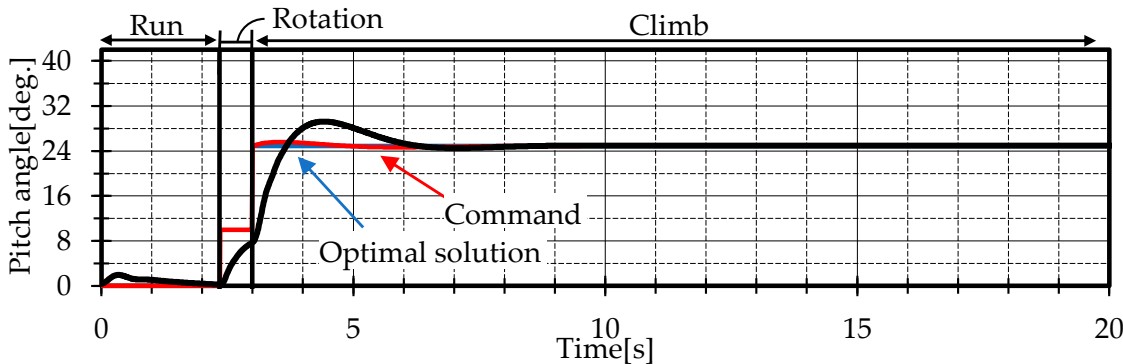

**Figure 11.** Simulated profile of pitch angle.

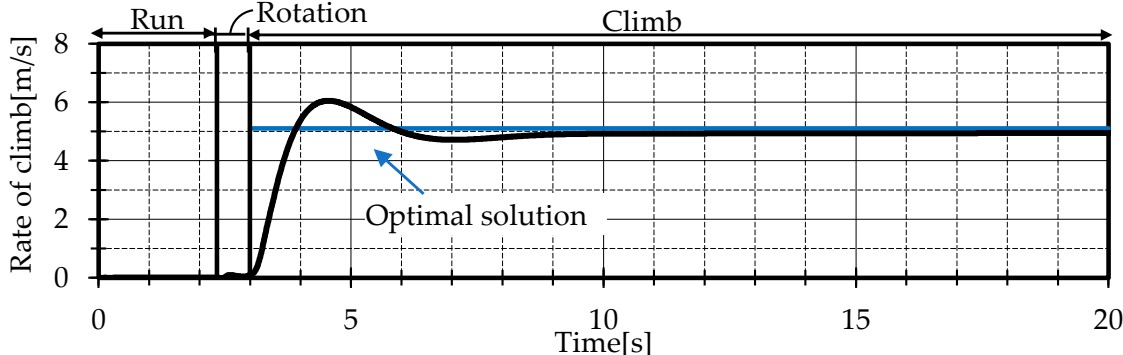

**Figure 12.** Simulated profile of rate of climb.

### 4. Flight Experiment

To confirm the validity of the takeoff at the maximum rate of climb, a flight experiment was carried out using the model airplane at the Shiraoi gliding port in Hokkaido, Japan. In this experiment, during the run, the elevator angle was controlled to be 3 degrees in the direction of pitch angle down so as to push the airplane to the runway, so as to improve the steering performance. Other than that, the same control system as in the simulation was used except for the PID parameters of the PID controller, which were determined by trial and error in the flight experiments.

#### 4.1. Judgement Conditions

In the actual environment, there exist many factors which have influence on flight performance gained by flight experiments. In order to confirm the validity of our method using the experiment, it is important to set in advance the judgement value of flight performances to determine whether the airplane is in steady climb. The sensor noises for airspeed and attitude angle in the ADS and INS used in the flight experiment are shown in Table 6. Ideally, during the climb, the pitch angle command is generated so that the airplane climbs at a constant airspeed without any deviation from the airspeed of the optimal method. However, as the airspeed measured by the ADS has a measurement noise, the pitch angle during the steady climb is influenced accordingly, as shown in Table 7. From the above, the criteria for judging the steady climb should be decided by using the variation range of the airspeed which is influenced by the sensor noise and balanced during the climb. Correspondingly to the variation of the airspeed, the pitch angle which realizes the airspeed varies. The criteria considering those variations are summarized in Table 8.

#### 4.2. Experimental Results

The results of the flight experiment are shown in Figures 13–18. The lateral position in Figure 14 was calculated from the latitude and longitude of the airplane measured using the INS. The altitude, pitch angle, and rate of climb of the airplane in Figures 15, 17 and 18

were measured using the INS. The airspeed of the airplane in Figure 16 was measured using ADS. The airplane took off for approximately 24 s from the start of the experiment and finally reached an altitude of 110 m, as shown in Figure 13. The following data are presented for the first 20 s after the start of the experiment to correspond with the simulation results of the previous section. Figure 14 shows that the airplane deviated laterally from the runway by a maximum of 8 m during takeoff, but approximately took off in a straight line along the runway centerline. Thus, during the flight experiment, the airplane only moved vertically. The airplane climbed to an altitude of about 80 m in 20 s after takeoff, as shown in Figure 15. The airspeed and pitch angle met the criteria for steady climb in 15 s after the start of the run, and the mean values of airspeed and pitch angle thereafter were 15.1 m/s and 22.0 degrees, respectively, although the pitch angle exceeded the command values by about 10 degrees just after the start of climb, as shown in Figures 16 and 17. The convergence value of the rate of climb was 5.7 m/s, as shown in Figure 18. The discrepancy between the optimal solution of 24.9 degrees and the experimental pitch angle of 22.0 degrees was caused by noise from sensors such as the ADS and uncertainties in the aerodynamic coefficient. Considering these factors, the pitch angle deviation of 2.9 degrees from the optimal angle in the flight experiment is acceptable, and the maximum rate of climb is achieved.

**Table 6.** Airspeed and pitch angle noise level.

| Flight Variable | Standard Deviation |
|---|---|
| Airspeed | 1.8 m/s |
| Pitch angle | 0.5 deg. |

**Table 7.** Pitch angle at which the airplane is balanced when the airspeed value deviates due to the noise amount.

| Flight Variable | Minimum | Nominal | Max |
|---|---|---|---|
| Airspeed discrepancy (true airspeed) | −1.8 m/s (13.3 m/s) | 0 m/s (15.1 m/s) | 1.8 m/s (16.9 m/s) |
| Pitch angle | 29.0 deg. | 24.9 deg. | 21.0 deg. |

**Table 8.** Range of variation criteria for steady climb.

| Flight Variable | Criteria |
|---|---|
| Airspeed | 3.6 m/s |
| Pitch angle | 8.0 deg. |

*4.3. Comparison of Flight Experiment and 6-DOF Flight Simulation Results*

The 6-DOF flight simulation did not take into account sensor noise or wind disturbances. As a result, the airspeed and pitch angle at steady climb matched the optimal solution, while the rate of climb differed from the optimal solution by 3.9%. On the other hand, in the flight experiment, the airspeed at steady climb matched the optimal solution, but the pitch angle did not match the optimal solution. In addition, the rate of climb differed from the optimal solution by 11.8%. These results are summarized in Table 9.

**Table 9.** Comparison of values by the optimal solution and flight verification.

| | Airspeed | Pitch Angle | Rate of Climb |
|---|---|---|---|
| Optimal solution | 15.1 m/s | 24.9 deg. | 5.1 m/s |
| Simulation | 15.1 m/s | 24.9 deg. | 4.9 m/s |
| Experiment | 15.1 m/s | 22.0 deg. | 5.7 m/s |

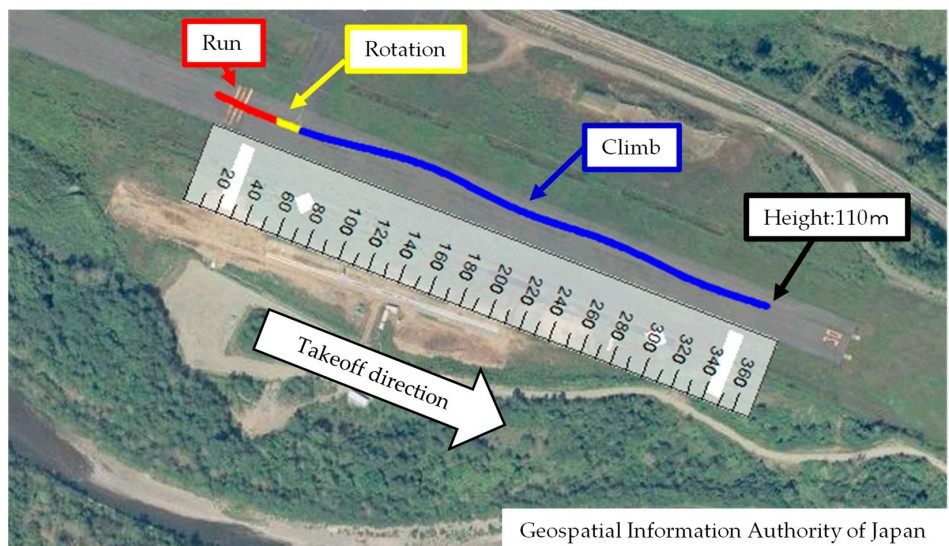

**Figure 13.** Experimental flight trajectory.

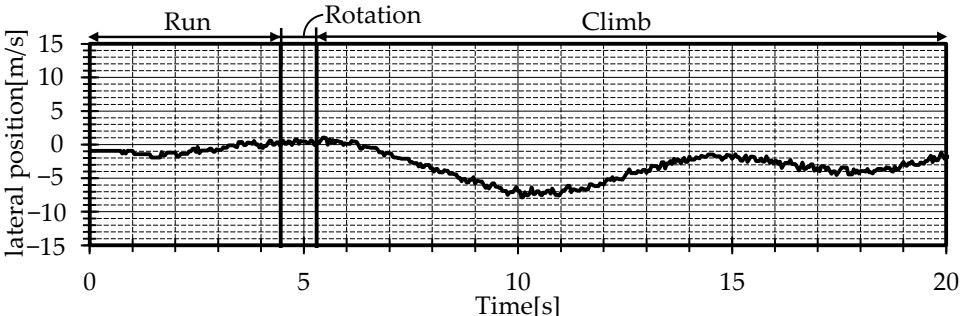

**Figure 14.** Experimental profile of lateral position.

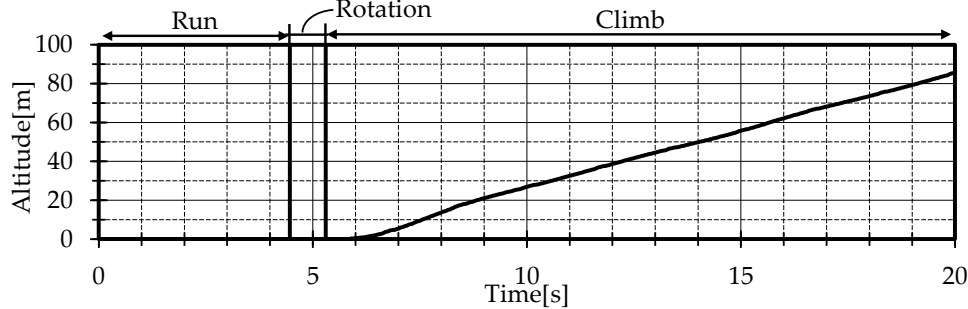

**Figure 15.** Experimental profile of altitude.

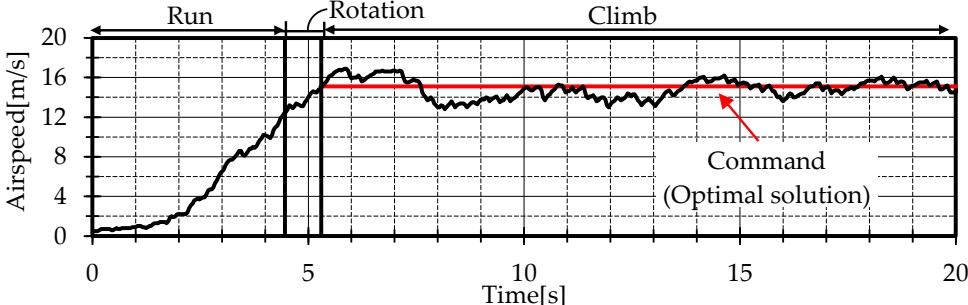

**Figure 16.** Experimental profile of airspeed.

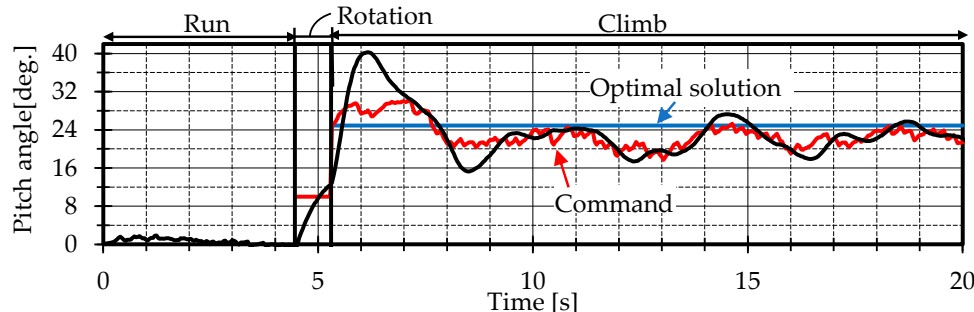

**Figure 17.** Experimental profile of pitch angle.

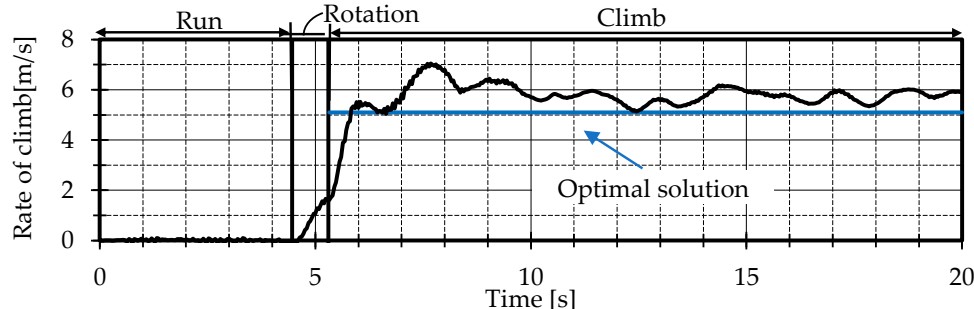

**Figure 18.** Experimental profile of rate of climb.

## 5. Conclusions

The characteristic of the propeller engine that thrust decreases with the increasing airspeed of the airplane was formulated and incorporated into the optimization problem to derive the maximum rate of climb of a fixed-wing UAV which is driven by a propeller engine. Subsequently, the takeoff at this rate was confirmed to be feasible through the 6-DOF flight simulation including the entire takeoff (run, rotation, and climb) Finally, flight experiments were conducted to confirm the validity of the rate. The experimental results showed that the rate of climb of the airplane deviated from the optimal solution, but this deviation was acceptable considering the error in the aerodynamic coefficient and the sensor noise. Thus, the validity of the maximum rate of climb was confirmed.

**Author Contributions:** Conceptualization, K.W.; validation, T.S.; resources, M.U.; writing—original draft, K.W.; writing—review and editing, M.U. and T.S.; supervision, M.U. All authors have read and agreed to the published version of the manuscript.

**Funding:** This research and the APC were funded by Aerospace Plane Research Center (APReC), Muroran Institute of Technology.

**Data Availability Statement:** Due to the fact that the research project of Oowashi of APReC is ongoing, no data could be shared.

**Conflicts of Interest:** The authors declare no conflicts of interest.

## Nomenclature

| | | | |
|---|---|---|---|
| $g$ | gravitational acceleration, m/s$^2$ | $c$ | chord length, m |
| $\rho$ | atmospheric density, kg/m$^3$ | $e$ | Oswald efficiency number |
| $L$ | lift, N | $AR$ | aspect ratio |
| $D$ | drag, N | $S$ | wing area, m$^2$ |
| $W$ | weight, N | $J$ | evaluation function |
| $\alpha$ | angle of attack, rad | $H$ | Hamiltonian |
| $\gamma$ | path angle, rad | $\lambda_1, \lambda_2$ | adjoint variable |
| $C_{L0}$ | coefficient of lift | $V$ | airspeed, m/s |

| $C_{L\alpha}$ | lift per unit AoA, 1/rad | $V_R$ | rotation speed, m/s |
|---|---|---|---|
| $C_{D0}$ | coefficient of drag | $V_2$ | takeoff safety speed, m/s |
| $\alpha_{stall}$ | stall angle | T | thrust, N |
| m | mass, kg | $\theta$ | pitch angle, rad |
| $\delta$ | angle of aileron, elevator, steer | $\phi$ | roll angle, rad |
| $\psi$ | azimuth angle, rad | | |
| Subscripts | | | |
| e | elevator | opt | optimal solution |
| a | aileron | cmd | command |
| s | steer | | |

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
