# Peer review of "Derivation and Flight Test Validation of Maximum Rate of Climb during Takeoff for Fixed-Wing UAV Driven by Propeller Engine"

_aerospace, doi:10.3390/aerospace11030233_

Round 1

Reviewer 1 Report (New Reviewer)

Comments and Suggestions for Authors

Please find my commnets attached.

Author Response

Thank you for taking the time to review this manuscript. Please find our detailed response and corresponding corrections and revisions in the attached file.

Reviewer 2 Report (New Reviewer)

Comments and Suggestions for Authors

The paper analyzes the optimal climb process of a fixed-wing airplane, it is interesting. However, there are some questions:

(1) The 2.2 Application of optimization problems to climbing airplane are not very rich, it is the most important about the theory and fundamentals in the paper. However, the it is lack of innovation, and the Hamiltonian (7) satisfies Equations 125 (8) through (12), it is better to provide a more detailed explanation.

(2) Rotation control is completed through controlling the elevator. What is the control input from the rotation phase to the climb phase? And whether there is the possibility that the aircraft has already lifted off the ground due to the ground effect when rotating at 12.5m/s and less than 15.1m/s? When the altitude is greater than the height of the affected area by the ground effect, it may cause the aircraft to descend.

(3) There is a situation when the speed does not reach the desired speed and the airplane leaves the ground while the pitch angle is kept to be 0 degrees in line 198. The desired pitch angle can be a negative degree smaller than the angle of the runway to avoid the aircraft leaving the runway. the elevator angle is controlled to be a fixed angle to make the airplane bow down.

(4) In line 200 of the paper, the block diagram of the rotation control system in Figure 6. The ailerons are controlled to keep the airplane horizontal. Has the airplane not lifted off the runway at this time? If the runway is not horizontal (the aircraft deviates from the runway side), Control the ailerons., it may cause instability of the aircraft and even wingtip touch the ground.

(5) It is necessary to clearly describe the lateral-directional and longitudinal control of the three processes run, rotation and climb.

(6) Is it necessary to maintain the airplane on the centerline of the runway after it leaves the runway.

Comments on the Quality of English Language

(1) The format and writing, the first letter of the word in the title of the reference paper capitalized uniformly.

Author Response

Thank you for taking the time to review this manuscript. Please find our detailed response and corresponding corrections and revisions in the attached file.

Round 2

Reviewer 1 Report (New Reviewer)

Comments and Suggestions for Authors

Dear authors,

After careful consideration of your revised manuscript, I would like to thank you for addressing all the suggested comments.

This manuscript is a resubmission of an earlier submission. The following is a list of the peer review reports and author responses from that submission.

Round 1

Reviewer 1 Report

Comments and Suggestions for Authors

(1)     Different pitch angles result in different thrust characteristics. The method proposed in this paper assumes that the thrust is only related to velocity, which seems unreasonable.

(2)     It is suggested to consider the matching between the engine and propeller.

Reviewer 2 Report

Comments and Suggestions for Authors

The paper presents the optimization of a simple 2D aircraft model for rate of climb. The design variables were trim states.The authors used experimental data curve fit for the thrust model.

The work is fine but there is nothing novel in there for it to be published in archival journal. This is something we expect our undergraduate students to do for a senior thesis. The work is good for a senior thesis but not for publication.

There is no significant novel technical content.

Comments on the Quality of English Language

The English is OK with some grammatical errors. I believe the authors meant Lagrangian instead of Hamiltonian.